# Efficacy of nonviral gene transfer of human hepatocyte growth factor (HGF) against ischemic-reperfusion nerve injury in rats

Toyokazu Tsuchihara[1,2¤a], Hitoshi Nukada[2,3]*, Kuniaki Nakanishi[4¤b], Ryuichi Morishita[5], Masatoshi Amako[1], Hiroshi Arino[1¤c], Koichi Nemoto[1¤d], Kazuhiro Chiba[1]

1 Department of Orthopedic Surgery, National Defense Medical College, Tokorozawa, Japan, 2 Department of Medicine, University of Otago Medical School, Dunedin, New Zealand, 3 Department of Exploratory Medicine on Nature, Life, and Man, Toho University Medical School, Chiba, Japan, 4 Department of Laboratory Medicine, National Defense Medical College Hospital, Tokorozawa, Japan, 5 Department of Clinical Gene Therapy, Osaka University Graduate School of Medicine, Suita, Japan

¤a Current address: Tsuchihara Orthopaedic Clinic, Yokohama, Japan
¤b Current address: Department of Pathology, KKR-Tachikawa Hospital, Tachikawa, Japan
¤c Current address: Department of Orthopedic Surgery, Ota Memorial Hospital, Ota, Japan
¤d Current address: Department of Orthopaedics, Eijinkai Iruma Heart Hospital, Iruma, Japan
* hitoshi.nukada@med.toho-u.ac.jp

**Data Availability Statement:** All relevant data are within the manuscript and its Supporting Information files.

## Abstract

Ischemic neuropathy is common in subjects with critical limb ischemia, frequently causing chronic neuropathic pain. However, neuropathic pain caused by ischemia is hard to control despite the restoration of an adequate blood flow. Here, we used a rat model of ischemic-reperfusion nerve injury (IRI) to investigate possible effects of hepatocyte growth factor (HGF) against ischemic neuropathy. Hemagglutinating virus of Japan (HVJ) liposomes containing plasmids encoded with HGF was delivered into the peripheral nervous system by retrograde axonal transport following its repeated injections into the tibialis anterior muscle in the right hindlimb. First HGF gene transfer was done immediately after IRI, and repeated at 1, 2 and 3 weeks later. Rats with IRI exhibited pronounced mechanical allodynia and thermal hyperalgesia, decreased blood flow and skin temperature, and lowered thresholds of plantar stimuli in the hind paw. These were all significantly improved by HGF gene transfer, as also were sciatic nerve conduction velocity and muscle action potential amplitudes. Histologically, HGF gene transfer resulted in a significant increase of endoneurial microvessels in sciatic and tibial nerves and promoted nerve regeneration which were confirmed by morphometric analysis. Neovascularization was observed in the contralateral side of peripheral nerves as well. In addition, IRI elevated mRNA levels of P2X3 and P2Y1 receptors, and transient receptor potential vanilloid receptor subtype 1 (TRPV1) in sciatic nerves, dorsal root ganglia and spinal cord, and these elevated levels were inhibited by HGF gene transfer. In conclusion, HGF gene transfer is a potent candidate for treatment of acute ischemic neuropathy caused by reperfusion injury, because of robust angiogenesis and enhanced nerve regeneration.

**Funding:** Grants-in-Aid for Scientific Research from the Ministry of Education, Culture, Sports, Science and Technology of Japan to Toyokazu Tsuchihara (No. 20591037 and 24591307). https:// www.jsps.go.jp/index.html https://www.mext.go. jp/a_menu/shinkou/hojyo/main5_a5.htm The funders had no role in study design, data collection and analysis, decision to publish, or preparation of the manuscript.

**Competing interests:** The authors have declared that no competing interest exists.

**Abbreviations:** HGF, Hepatocyte growth factor; IRI, ischemic-reperfusion nerve injury; CMAP, compound muscle action potentials; MNCV, motor nerve conduction velocity; TRPV1, transient receptor potential vanilloid receptor subtype 1; HVJ, hemagglutinating virus of Japan; CPT, current perception threshold; ANOVA, analysis of variance; GAPDH, glyceraldehyde 3-phosphate dehydrogenase; DRG, dorsal root ganglia; CCI, chronic constriction injury.

## Introduction

Ischemic neuropathy is common among patients with lower extremity peripheral artery disease [1,2]. The incidence of critical limb ischemia caused by peripheral artery disease has been estimated to be approximately 500–1000 per million per year [3–6]. Clinical neurological deficits have been reported to occur in 22–88% of critical limb ischemia cases [7]. Sensory symptoms, such as numbness, painful paresthesia and burning [8,9], may last for a long time in spite of reestablishing an adequate blood flow [10,11]. The incidence of ischemic neuropathy may be expected to increase in line with the current and projected increases in the elderly and obese populations. However, effective therapies for ischemic neuropathy are not established.

Gene transfer via retrograde axonal transport is a possible way of expressing identified transgenes at targeted locations within the nervous system. Although the molecular mechanism of retrograde transport of HGF is not fully understood, we have achieved successful gene transfer into the rat peripheral nerves via retrograde axonal transport from intramuscular injections with either (a) with hemagglutinating virus of Japan (HVJ)-liposomes containing plasmids encoded with luciferase or ß-galactosidase, or (b) with HVJ envelope encoded with hepatocyte growth factor (HGF) [12,13].

HGF, which was initially purified from rat serum [14,15], has powerful neurotrophic and angiogenic effects [16–21]. Within the nervous system, HGF is trophic for peripheral sensory and motor neurons and enhances axonal outgrowth [16,22–28]. We have demonstrated effective control of neuropathic pain by repeated intramuscular injections of human HGF in rat models of CCI [29] which was confirmed later [30]. HGF is also a powerful growth factor of endothelial cells and has been investigated for the treatment of ischemic diseases [19,31]. Therapeutic angiogenesis with HGF has been reported in patients with critical limb ischemia [32–35].

In the search for an agent to improve the symptoms of ischemic neuropathy, HGF is a potent candidate because of its powerful neurotrophic and angiogenic properties. We have investigated whether nonviral retrograde gene transfer of HGF might improve ischemic neuropathy using the rat model of ischemic-reperfusion nerve injury (IRI).

## Materials and methods

### Rats

All animal procedures were carried out in compliance with the Guide for the Care and Use of Laboratory Animals of the National Institutes of Health. The protocol was approved by the Committee on the Ethics of Animal Experiments of the National Defense Medical College (Protocol Number: 07021). One hundred-eight male Wistar rats, weighing 220–280 g, age 8–9 weeks were analyzed in this study. Animals were housed in a temperature-controlled room with a 12-hour light-dark cycle. During the procedure, all efforts were made to minimize suffering, to minimize the number of animals, to utilize alternative techniques, if available, to relieve pain and /or distress as necessary, and to increase the level of monitoring if rats exhibit the sign of morbidity. Rats were euthanized by $CO_2$ inhalation followed by cervical dislocation.

### Study design

The method used to induce IRI was as described by Nukada and McMorran [36]. Briefly, in rats under sodium pentobarbital anesthesia the major arteries supplying the right hindlimb; abdominal aorta, common iliac, femoral, and superficial circumflex iliac arteries, were exposed and occluded for 4 h using microvascular clips (TKS- 1, 40g; Bear Medic Corporation, Ichikawa, Japan), and then reperfused.

Immediately after reperfusion, HVJ envelopes (100 μl) containing 100 μg human HGF plasmid DNA, were injected into the proximal one-third of the tibialis anterior muscle of the right hindlimb over a few minutes via a 27-gauge needle. HGF gene transfer was repeated at 1, 2 and 3 weeks after IRI. Rats were divided randomly into 4 groups as follows: (1) IRI with HGF gene transfer (IRI+HGF group), (2) IRI without HGF gene transfer (IRI group), (3) similar surgery to expose arteries but no IRI (sham group), and (4) no surgical procedures (control group).

## Construction of plasmid DNA and preparation of the HVJ envelope-complex vector

To produce an HGF expression vector, human HGF cDNA (2.2 kb) was inserted between the NotI sites of a pcDNA 3.1 (−) vector. The HVJ envelope-complex vector was prepared using an HVJ envelope vector kit (GenomOne-Neo; Ishihara Sangyo, Osaka, Japan) as described previously [13].

In our previous studies, we have demonstrated that HGF expression in ipsilateral sciatic nerve and dorsal root ganglia (DRG) was high on day 3 and decline on Day 7 after the first gene transfer, and was high again day 3 after the second gene transfer [29]. We have also confirmed the specific primers we used against rat HGF mRNA did not cross-react with human HGF mRNA [29].

## Behavioral studies

Mechanical allodynia and thermal hyperalgesia of the right hind paw were measured at 14 defined time-points until 8 weeks post-operation (1, 2, 3, 5, 7, 10, 14, 17, 21, 24 days, and 4, 5, 6, 7 weeks). Methods have been reported previously [29,37,38]. In brief, mechanical allodynia was determined by quantifying the withdrawal threshold of the soles of right hind paw in response to Von Frey hairs. Von Frey hairs were applied perpendicularly to the plantar surface of the hind paw, sufficient force being used to bend the filament. Brisk withdrawal or paw flinching was taken as a positive response.

Thermal hyperalgesia was assessed using a Hargreaves apparatus (Plantar Test; Ugo Basile, Italy). A radiant heat stimulus was applied to the midplantar area of the hind paw, and the time between initial heat and paw withdrawal was recorded. Each testing for mechanical allodynia and thermal hyperalgesia was repeated five times, and the mean value was used.

## Laser Doppler flux and thermographic studies of the hind paw

Blood flow and thermographic measurements in the hind paw were measured at baseline and 1, 3, 6 and 8 weeks after IRI, as in our previous study [29]. Blood flow was using a laser Doppler flowmeter (PIM II: Perimed AB, Järfälli, Sweden), and thermographic measurements was performed using a TH9100 ML camera (NEC Sanei, Tokyo, Japan).

## Neurometer measurement of current stimulus threshold

The stimulus thresholds were measured using a Neurometer according to the method of Kiso et al. [39]. Briefly, a small electrode (GT100-30; Neurotron Inc., Baltimore, USA) for stimulation was attached to the plantar surface of the hind paw. A skin patch dispersion electrode (TE174D; Fukuda Denshi, Tokyo, Japan) was attached to the lower back of the rat. In rats lightly restrained using Ballman cages (Natsume, Tokyo, Japan), transcutaneous nerve stimuli —each comprising three sine-wave pulses at 5, 250, or 2,000 Hz—were applied to the plantar surface, using an animal response test mode of the Neurometer Current Perception Threshold (CPT) (Neurotron Inc., Baltimore, MD) [40]. Delivery of sinusoidal electrical stimuli at 5, 250,

or 2,000 Hz allows evaluation of responses primarily in small unmyelinated (C), small myelinated (Aδ), and large myelinated (Aβ) fibers, respectively [41–45]. The minimum current intensity (mA) at which each rat vocalized was defined as the CPT. The data presented herein were obtained from three consecutive measurements, and the CRT was measured before surgery and at 1, 3, 6 and 8 weeks after surgery.

### Nerve conduction study

Rat was on a heating pad, and the right hindlimb skin temperature was kept 35 ± 1ºC. The right sciatic nerve was exposed from the sciatic notch to the knee under general anesthesia with sodium pentobarbital. The nerve was stimulated using hook electrodes at the sciatic notch and knee. Recording needle electrodes were placed in the 1st and 4th interosseus muscles of the foot. Motor nerve conduction velocity (MNCV) and compound motor action potential (CMAP) were recorded using Neuropack instrumentation (Nihon Koden, Tokyo, Japan) as described previously [46]. Nerve conduction studies of the right sciatic nerve were carried out before surgery, and at 1, 3, 6, and 8 weeks after surgery.

### Histological studies of the sciatic nerve

After measurements of MNCV and CMAP at 3, 6 and 8 weeks post-surgery, bilateral sciatic and tibial nerves were taken from the upper-thigh to ankle level under sodium pentobarbital anesthesia (n = 6 at each time-point for each group) [36]. All specimens were fixed overnight in 2.5% glutaraldehyde in 0.1 M phosphate buffer solution (pH 7.4). Subsequently, they were postfixed with 1% osmium tetroxide, dehydrated, and cut into consecutive 2–3 mm blocks before embedding in epoxy resin. Semi-thin sections (1 μm in thickness) were stained with methylene blue. Ultra-thin sections were stained with EM stainer followed by lead citrate and examined with a transmission electron microscope (H7600, Hitachi, Tokyo, Japan). Morphometric analysis of the number and diameter of myelinated nerve fibers and the number of endoneurial microvessels in sciatic and tibial nerves, was determined using a NIH image analysis.

### RT-PCR

The mRNA expressions for P2X2, P2X3, and P2X4 receptors, and TRPV1 (transient receptor potential vanilloid receptor subtype 1) in right sciatic nerve, right DRG (L5), and spinal cord were examined at 1 and 3 weeks post-operatively by semiquantitative RT-PCR as described previously [29]. Sequence information used in the current study is listed in S1 Table.

### Statistical analysis

Values for each parameter are expressed as means ± SD. Fisher's protected least-significant-difference test was applied to the data when significant F-ratios were obtained in an analysis of variance (ANOVA), and significance was set at $p < 0.05$. The statistical significance of the difference between curves representing repeated measures of behavioral function (allodynia or hyperalgesia) was assessed using a repeated measures analysis (SPSS Inc., Chicago, IL, USA).

## Results

### Mechanical and thermal thresholds

On day 1 post-surgery, right hind paw withdrawal threshold using Von Frey hairs in both IRI +HGF and IRI groups was significantly decreased when compared with pre-surgical levels, and was also significantly less than in control and sham groups (Fig 1A). The mechanical

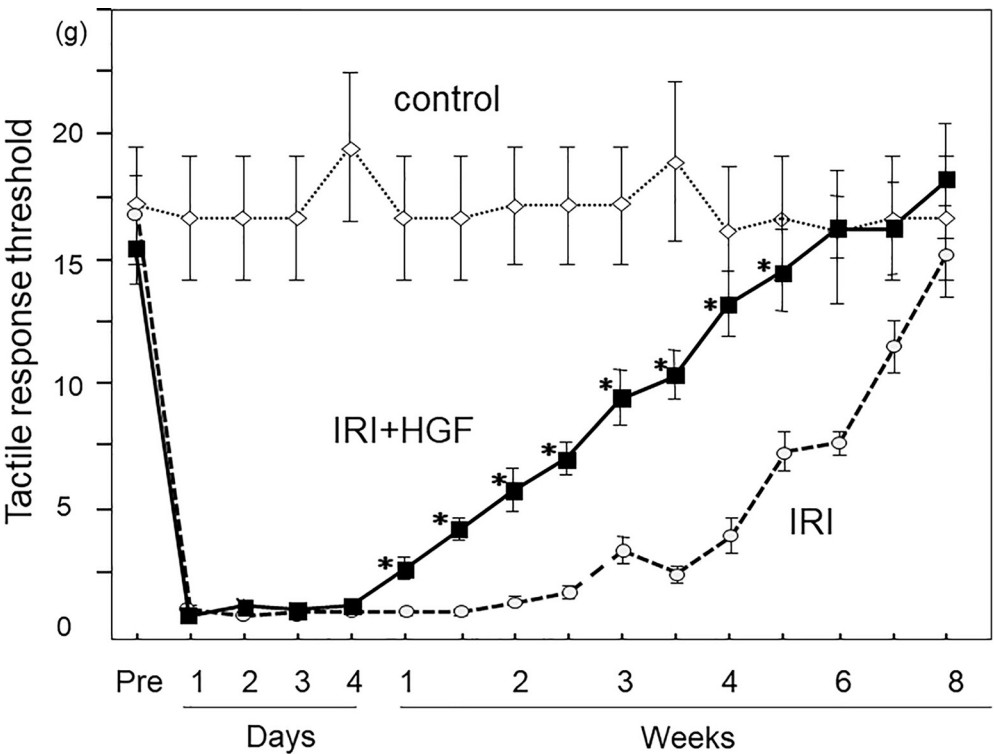

Time after the first gene transfer

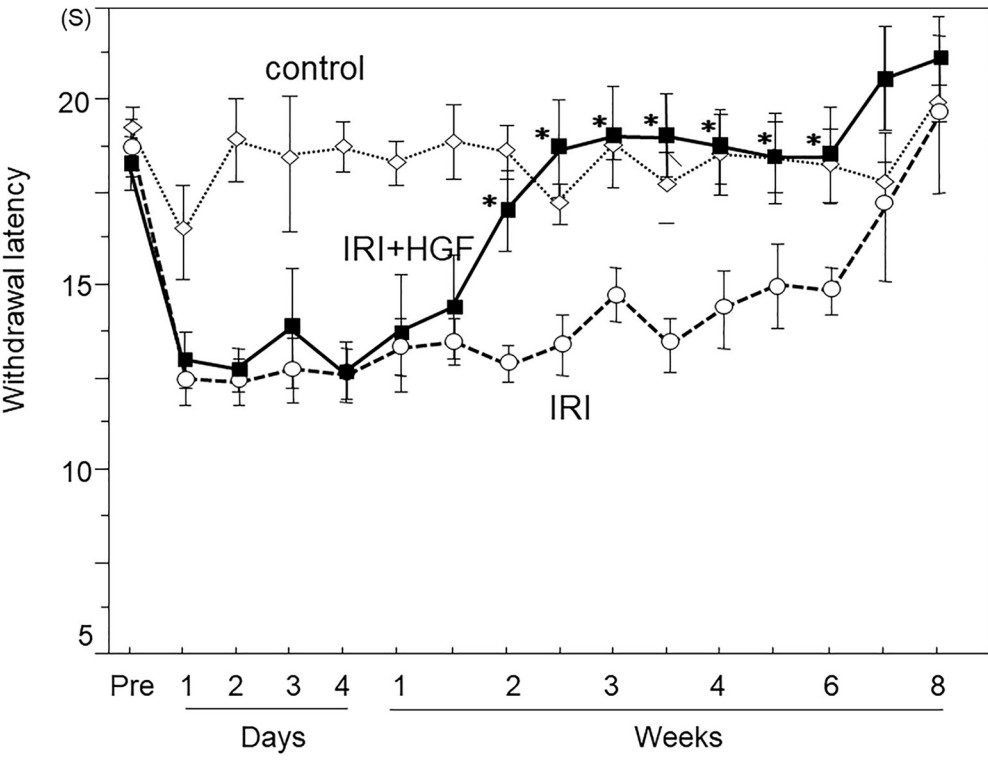

Time after the first gene transfer

**Fig 1. Behavioral studies: Mechanical allodynia and thermal hyperalgesia.** Time-course data for tactile-response threshold using Von Frey hairs **(A)** and withdrawal latency to infrared heat stimulus **(B)** of the right hind paw in IRI (○ broken line), IRI+HGF (■ solid line), and control (◇ dotted line) groups. Data are expressed as mean ± SD (n = 12 for each group at each time point). $^*p < 0.01$: IRI+HGF $vs.$ IRI.

threshold in IRI+HGF group became significantly greater than in IRI group on week 1 post-surgery, and thereafter increased with time (Fig 1A). The recovery of threshold in IRI+HGF group was faster than in IRI group, and at week 5 post-surgery, threshold in IRI+HGF group became not significantly different from those in controls. The mechanical threshold of left hind paw in IRI+HGF and IRI groups was reduced, although statistically not significant. Raw data of these thresholds of bilateral hind paws in IRI+HGF, IRI, control and sham groups were shown in S2 Table.

Thermal threshold was significantly reduced at day 1 post-surgery in both IRI and IRI +HGF groups when compared with those in control and sham groups (Fig 1B). However, the recovery of threshold was much faster in IRI+HGF group than in IRI group. At week 2 post-surgery, there was no significant differences of thresholds between IRI+HGF group and controls. Thermal threshold in IRI+HGF group was significantly greater from week 2 to 6 post-surgery than in IRI group (Fig 1B). There were no significant changes of behavioral studies in the control and sham groups.

## Laser Doppler flux and thermographic studies

Laser Doppler flux of the right hind paw in IRI+HGF group was not significantly different at all time-points from those in controls. Blood flow in IRI group was significantly decreased at weeks 1 and 3 post-surgery than those in IRI+HGF group (Fig 2). Blood flow in the left hind paw was not significantly different between experimental groups (Fig 2).

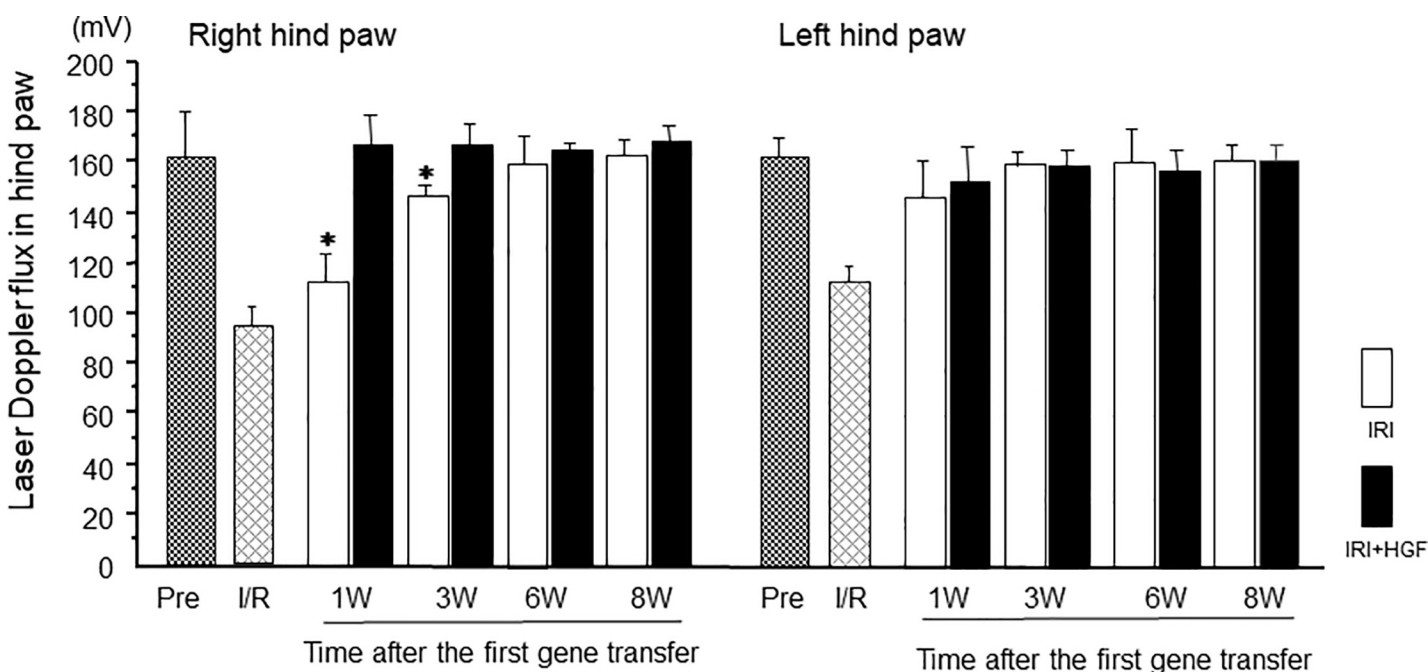

**Fig 2. Skin blood flow by laser Doppler flux.** Time-course data for laser Doppler flux of right and left hind paws in IRI (open bars) and IRI+HGF (solid bars) groups. Pre (fine checkered bars): pre-surgery. I/R (coarse checkered bars): during ischemic-reperfusion surgery. Data are expressed as mean ± SD (n = 6 for each group at each time point). $^*p < 0.01$: IRI+HGF $vs.$ IRI.

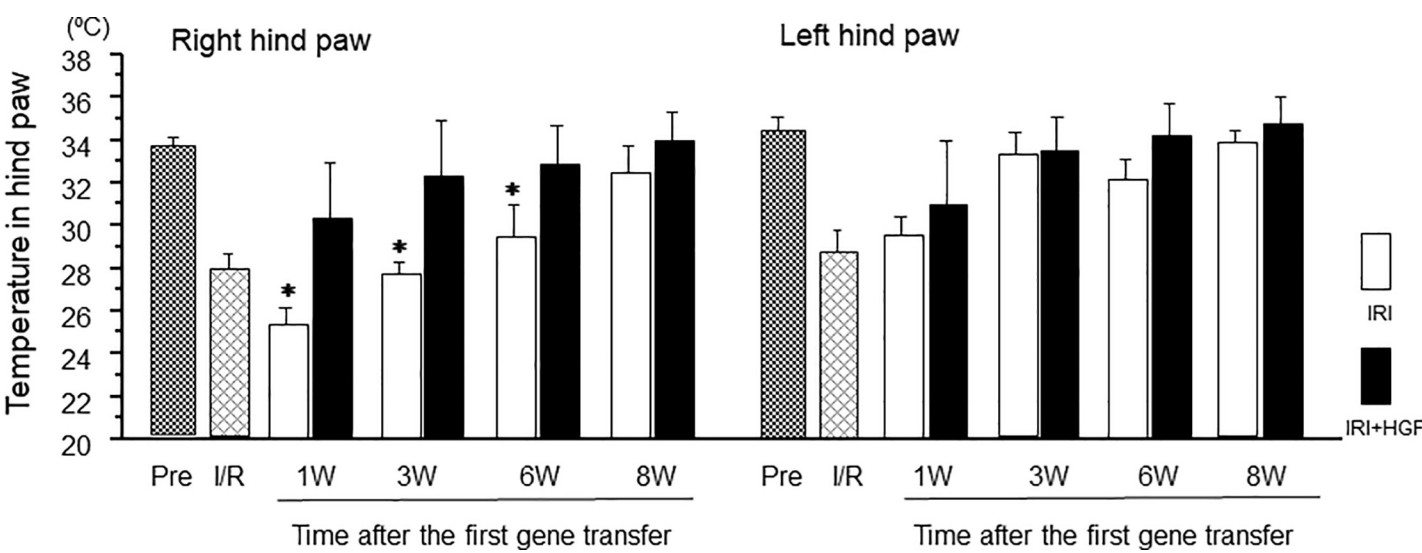

**Fig 3. Skin temperature by thermographic measurements.** Time-course data for thermographic skin temperature of right and left hind paws in IRI (open bars) and IRI+HGF (solid bars) groups. Pre (fine checkered bars): pre-surgery. I/R: (coarse checkered bars) during ischemic-reperfusion surgery. Data are expressed as mean ± SD (n = 6 for each group at each time point). *$p < 0.01$: IRI+HGF *vs*. IRI.

Skin temperature of the right hind paw in IRI+HGF group was significantly greater than in IRI group at weeks 1, 3, and 6 post-surgery, although it was reduced in both IRI+HGF and IRI groups when compared with pre-surgical levels (Fig 3). Skin temperature in the left hind paw of both IRI+HGF and IRI groups was significantly reduced at week 1 when compared with pre-surgical level (Fig 3). In control and sham groups, there were no significant changes of blood flow and thermographic measurements.

## Neurometer measurement of current stimulus threshold

Regardless of whether stimuli were delivered at 5, 250, or 2,000 Hz, the threshold at the right hind paw was significantly reduced in both IRI+HGF and IRI groups when compared with pre-surgical level. However, the threshold in IRI+HGF group was significantly greater than in IRI group at 1, 3, 6 and 8 post-surgery (Fig 4). In the left hind paw, thresholds at all current stimuli were lowered in both IRI+HGF and IRI groups at 1 week post-surgery, and at 2,000 Hz stimulus, thresholds in IRI+HGF group were significantly greater than in IRI group at weeks 1 and 3 (S1 Fig). In control and sham groups, no significant changes have been observed.

## Nerve conduction study

MNCVs and CMAP of the right sciatic nerves were significantly reduced in both IRI+HGF and IRI groups when compared with pre-surgical levels (Fig 5). However, MNCVs in IRI +HGF group were significantly faster than in IRI group at weeks 3 and 6 post-surgery (Fig 5A). CMAPs amplitude in IRI+HGF group was significantly greater than in IRI group (Fig 5B) at week 3. Recovery of slowed MNCVs and lowered CMAP amplitudes were faster in IRI +HGF group than in IRI group, correlated with the recovery of foot and toe drop.

## Histological study of the sciatic nerve

Semi-thin transverse sections at the lower-thigh level of right sciatic nerve obtained at week 3 post-surgery revealed a loss of large myelinated nerve fibers in both IRI+HGF and IRI groups,

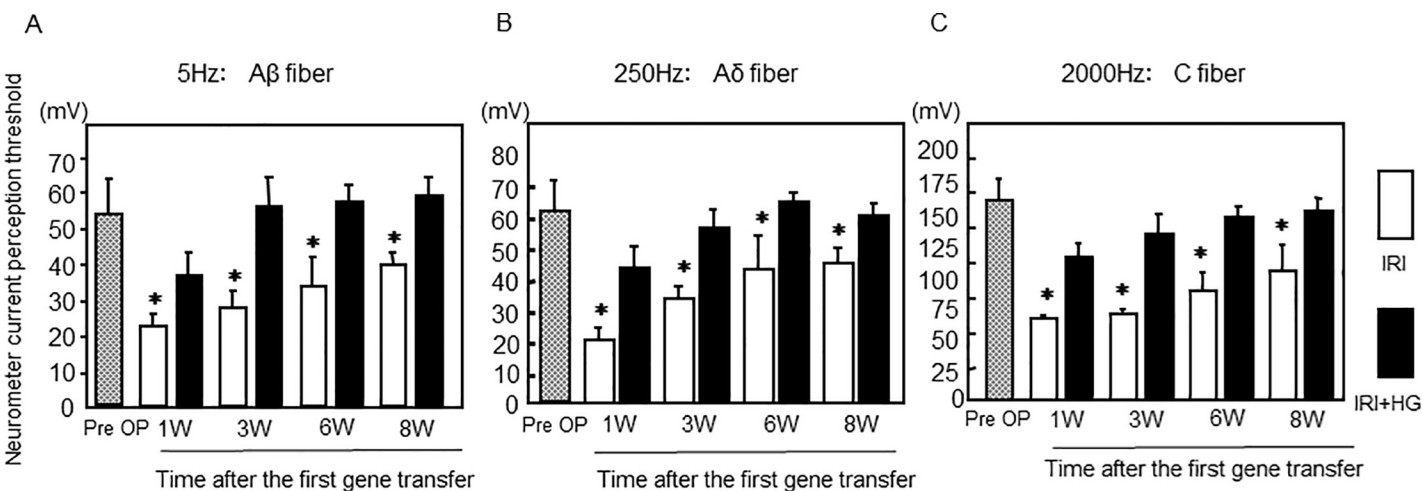

**Fig 4. Neurometer current perception threshold.** Neurometer measurements of stimulus threshold at 5 (**A**), 250 (**B**) and 2,000 (**C**) Hz to the plantar surface of the right hind paw in IRI (open bars) and IRI+HGF (solid bars) groups, and pre-surgical levels (Pre OP, checkered bars). Data are expressed as mean ± SD (n = 6 for each group at each time point). *$p < 0.01$: IRI+HGF *vs.* IRI.

although the number of large myelinated nerve fibers was apparently greater in IRI+HGF group than in IRI group (Fig 6). Small myelinated nerve fibers observed in both groups were mainly regenerating nerve fibers. Nerve fibers with disproportionately thin myelin relative to axon area were observed (S2 Fig), suggesting regenerating nerve fibers. Nerve fibers with axonal degeneration were scattered in both IRI and IRI+HGF groups, although more prominent in IRI group (Fig 6). Unmyelinated nerve fibers were apparently normal under the electron microscopy (S2 Fig). Endoneurial edema and vascular changes were not apparent.

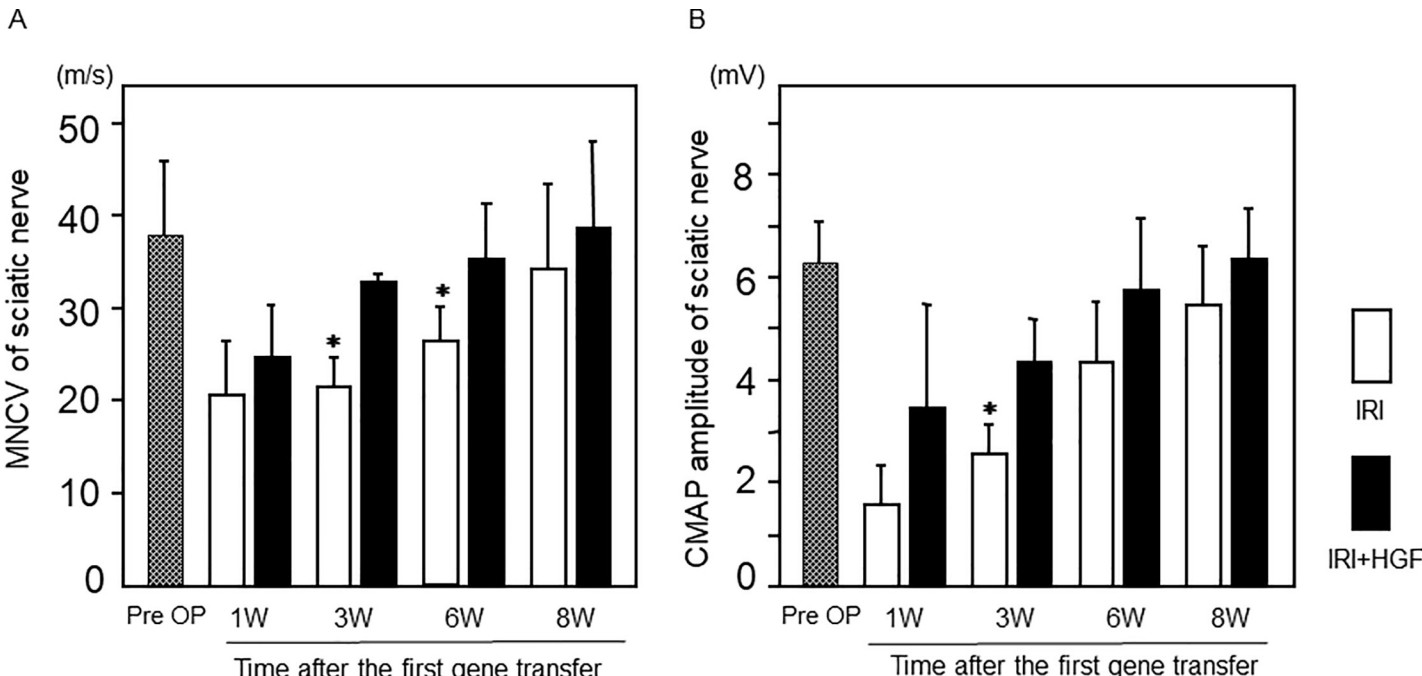

**Fig 5. Nerve conduction studies of sciatic nerves.** MNCV (**A**) and CMAP (**B**) of the right sciatic nerve in IRI (open bars) and IRI+HGF (solid bars) groups, and pre-surgical levels (Pre OP, checkered bars). Data are expressed as mean ± SD (n = 6 for each group at each time point). *$p < 0.01$: IRI+HGF *vs.* IRI.

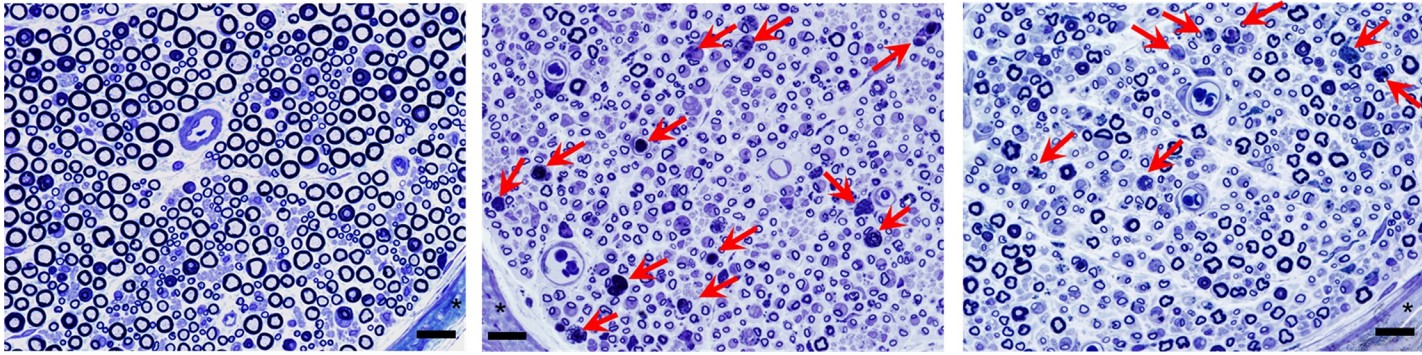

**Fig 6. Histological evaluation of sciatic nerves.** Semi-thin transverse sections at the lower thigh level of the right sciatic nerve at 3 weeks post-surgery in sham (A), IRI (B) and IRI+HGF (C) groups. A loss of large myelinated nerve fibers and the presence of small myelinated nerve fibers were prominent in both IRI and IRI+HGF groups. Nerve fibers with axonal degeneration (arrows) were more noticeable in IRI group when compared with IRI+HGF group. *perineurium. Bars = 20μm.

Morphometric analysis of myelinated nerve fibers at the lower-thigh level of right sciatic nerve at 3weeks after surgery revealed that the number of myelinated nerve fibers, and mean diameter and area of myelinated nerve axons were significantly less in both IRI+HGF and IRI groups than in controls (Table 1). However, axon area and diameter were significantly greater in IRI+HGF group that in IRI group (Table 1). The diameter histogram of myelinated nerve fibers confirmed an increased number of small myelinated nerve fibers in both IRI+HHGF and IRI groups (S3 Fig). Peaks of axon diameters in both groups occurred at around 1.5μm, while the peak of controls was approximately 3.5μm.

In sciatic nerves at the lower-thigh level, fascicular area at 3 weeks after surgery was not significantly between experimental groups (Fig 7 and S3 Table). The density of endoneruial microvessels in both right and left sciatic nerves was significantly greater in IRI+HGF than in IRI, sham and control groups (Fig 7, S4 Fig and S3 Table). Transverse semi-thin sections of the contralateral (left) sciatic nerves at the upper-thigh level in IRI+HGF and IRI groups were showed in S5 Fig to highlight a contrast of the number of endoneurial microvessels. A significant increased density of endoneruial microvessels in IRI+HGF group was observed in bilateral tibial nerves as well (S6 Fig and S4 Table).

## RT-PCR

We detected mRNA expressions for P2X2, P2X3, P2X4, and P2Y1 receptors, and for TRPV1, in right sciatic nerve, right DRG (L5) and spinal cord at weeks 1 and 3 post-surgery. We calculated ratios for the expression levels of the various receptor mRNA to those of glyceraldehyde 3-phosphate dehydrogenase (GAPDH) mRNA. In sciatic nerves, ratios obtained for P2X3 receptor mRNA, P2Y1 receptor mRNA, and TRPV1 mRNA in IRI group were all significantly greater than in IRI+HGF and control groups at week 1 (Fig 8). The ratios of P2X3 and P2Y1

**Table 1. Morphometric parameters of myelinated nerve fibers at the lower-thigh level of right sciatic nerve at 3 weeks after ischemic-reperfusion injury.**

|  | IRI+HGF | IRI | control | $p^*$ |
|---|---|---|---|---|
| Number of myelinated nerve fibers | 1153 ± 109 | 1143 ± 86 | 1986 ± 155 | 0.34 |
| Axon diameter of myelinated nerve fibers (μm) | 2.01 ± 0.04 | 1.79 ± 0.01 | 3.16 ± 0.17 | 0.0063 |
| Axon area of myelinated nerve fibers (μm$^2$) | 5.05 ± 0.21 | 3.99 ± 0.14 | 11.51 ± 1.10 | 0.0036 |

Data are means ± SD

* IRI+HGF vs. IRI.

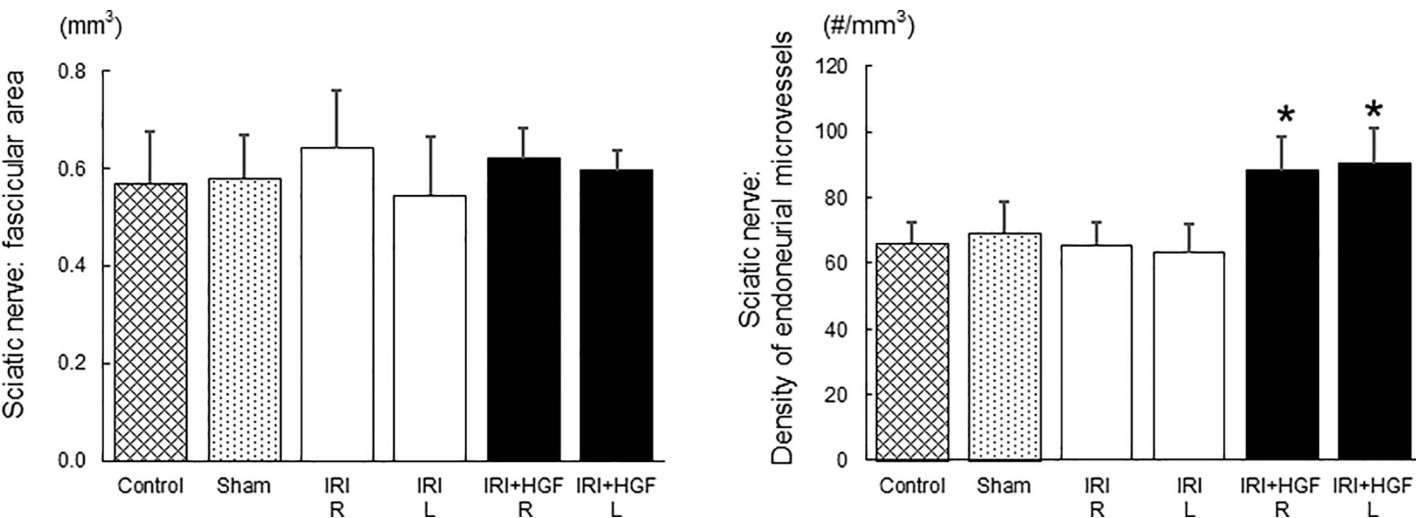

**Fig 7. Morphometry of fascicular area and endoneurial microvessels.** Fascicular area and the density of endoneurial microvessels at the lower-thigh level of bilateral sciatic nerves after 3 weeks post-surgery in IRI+HGF (solid bars), IRI (open bars), control (fine checkered bar) and sham (coarse checkered bar) groups. *$p < 0.01$: IRI +HGF *vs.* IRI, control and sham.

mRNAs at week 3 were not significantly different between three groups. Ratios obtained for TRPV1 mRNA at weeks 1 and 3 in IRI+HGF group were significantly less when compared with those in IRI group, although were significantly greater than in control group (Fig 8). No

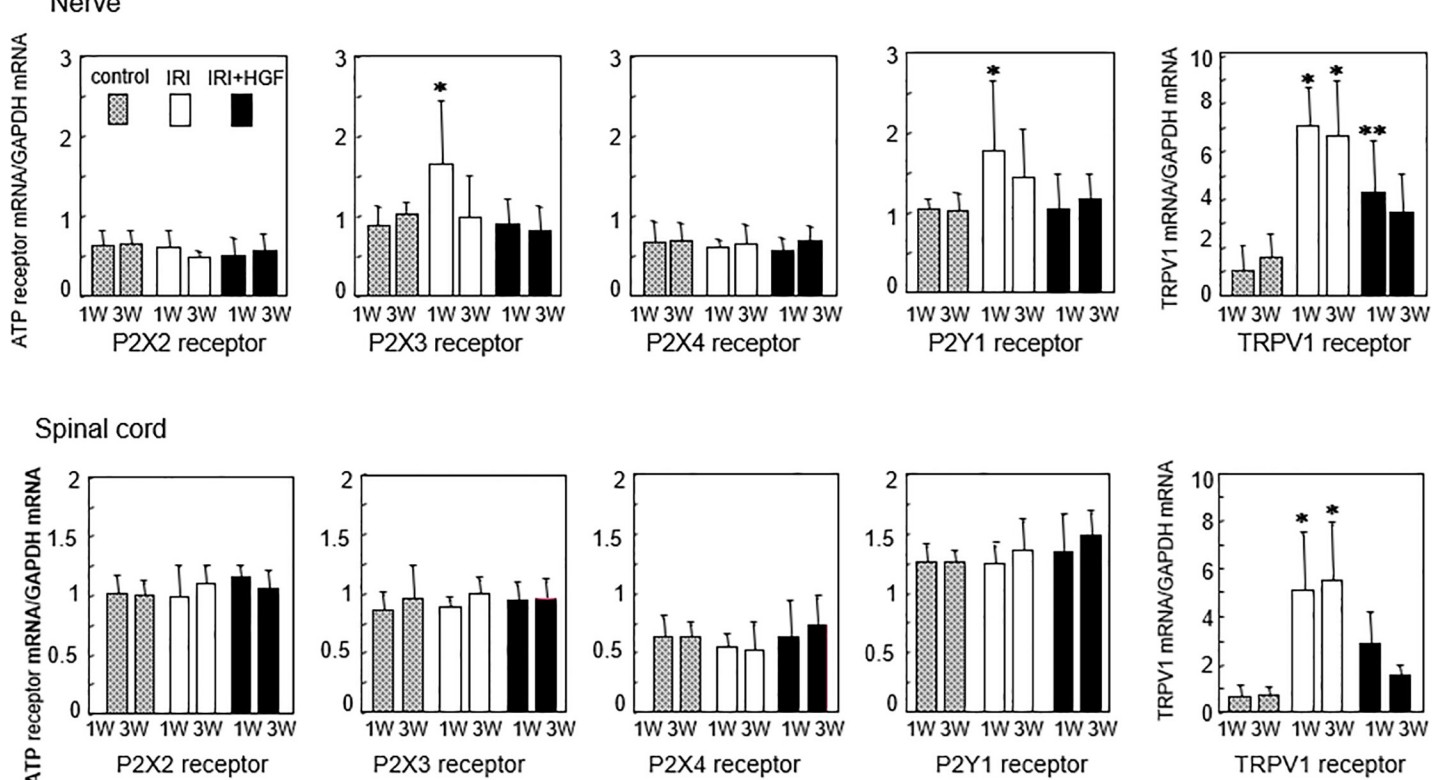

**Fig 8. RT-PCR studies of sciatic nerves and spinal cord.** mRNA expressions of P2X2, P2X3, P2X4, and P2Y1 receptors, and TRPV1 at weeks 1 and 3 post-surgery in IRI (open bars), IRI+HGF (solid bars), and control (checkered bars) groups. Data are expressed as mean ± SD (n = 6 for each group at each time point). *$p < 0.05$: IRI vs. IRI+HGF and control. **$p < 0.05$ IRI+HGF vs. IRI and control.

significant differences were found among 3 groups in the ratios obtained for P2X2 receptor mRNA and P2X4 receptor mRNA (Fig 8). In DRG, the mRNA levels were measured by mixing together all the specimens from 6 rats at each time-point, because of the small size of rat DRG. The ratios obtained for P2X3 and P2Y1 receptors and TRPV1 mRNAs in DRG at weeks 1 and 3 showed a similar trend as those of sciatic nerves; lower ratios in IRI+HGF than in IRI group (S7 Fig). In spinal cord, the ratio for TRPV1 mRNA in IRI+HGF group was significantly less when compared with those in IRI group at weeks 1 and 3, although mRNA expressions of P2X2, P2X3, P2X4 and P2Y1 receptors were not significantly different between three groups (Fig 8).

## Discussion

We have investigated whether nonviral retrograde gene transfer of HGF might improve symptoms of ischemic neuropathy using a rat model of IRI, and demonstrated that HGF gene transfer resulted in 1) significant improvement of mechanical allodynia and thermal hyperalgesia, decreased skin blood flow and temperature, lowered plantar thresholds of electrical stimuli in the hind paw, and sciatic nerve conduction parameters, 2) enhanced nerve regeneration in the sciatic nerve histologically, 3) significant increases of the density of endoneurial microvessels in bilateral sciatic and tibial nerves, and 4) reduction of the elevated levels of P2X3 and P2Y1 receptor mRNA and TRPV1 mRNA in the sciatic nerve. These results suggest that repeated retrograde gene transfer of HGF may improve symptoms of ischemic neuropathy.

Several well-defined underlying molecular mechanisms of retrograde transport of neurotrophic factors have been identified [47], and the efficacy of intramuscular injection over the calf muscle of HGF has been demonstrated. In our previous study, we demonstrated expression of human HGF-like immunoreactive protein in the rat sciatic nerve by repeated intramuscular injections of nonviral gene transfer of human HGF into the hindlimb muscle [29]. In fact, in the DRG and sciatic nerve at 3 days after the first and second HGF gene transfers, the levels of rat HGF mRNA and protein were increased 1.2- to 2.5-fold when compared with those in controls. We have also shown the efficacy of HGF upon neuropathic pain in a rat model of CCI [29]. Intramuscular injections of HGF have also been successful for treatment of models of streptozocin-induced diabetes [48] and crush injury [27]. Clinically, intramuscular administration of VM202, plasmid DNA expressing two isoforms of HGF, resulted in significant symptomatic relief in painful diabetic neuropathy [49]. In a double-blind, placebo-controlled study, Kessler at al. confirmed that repeated intramuscular injections of HGF are effective in reducing pain in patients with painful diabetic neuropathy [50].

There have been electrophysiological and histological studies of the ischemic neuropathy caused by IRI [36,51–53]. However, the mechanism causing sensory symptoms, e.g. numbness, paresthesia, burning and pain, is still not completely understood. In this study, we confirmed that mechanical allodynia and thermal hyperalgesia were evident within 1 day after IRI and persisted for at least 6 weeks. Repeated HGF gene transfer was successful in enhanced recover of these mechanical allodynia and thermal hyperalgesia after IRI. Using an animal response test mode of the Neurometer CPT, we also found that all nerve fibers including small unmyelinated (C), small myelinated (Aδ), and large myelinated (Aβ) fibers, were adversely affected after IRI and repeated HGF gene transfer led to an early full recovery to the pre-surgical level in all three fiber types.

By analyzing the morphometrical parameters of endoneurial microvessels and myelinated nerve fibers in sciatic nerves at 3 weeks after surgery, we found that 1) the number of endoneurial microvessels was significantly increased in IRI+HGF group than in IRI, control and sham groups, and 2) axon area and diameter of myelinated nerve fibers in IRI+HGF group

were significantly greater when compared with those in IRI group. Neovascularization by HGF gene transfer enhanced nerve regeneration, and an increased axon area in IRI+HGF group was due to the presence of intensified regenerating nerve fibers. Because ischemia causes delayed nerve regeneration [54,55], regenerating nerve fibers in IRI group were associated with a decrease in axon area.

A significant increased density of endoneurial vessels was observed in the contralateral sciatic and tibial nerves as well in IRI+HGF group. Because the transient ligation of abdominal aorta, at the level of distal to left iliolumbar artery and proximal to right illiolumar artery, during an ischemic surgery, transient ischemia at the mid-thigh level of sciatic nerve which is the watershed zone along the length of sciatic-tibial nerves [56,57], occurred in the contralateral hind limb [36,53]. Other results in the current study including behavioral studies, skin blood flow and temperature, and Neurometer measurements, also suggested mild ischemia in the contralateral side. Because s, the potent angiogenic property of HGF spread into the contralateral hind limb.

Levels of P2X3 and P2Y1 receptors, and TRPV1 mRNAs in sciatic nerves and DRG were elevated at 1 week after IRI, and these elevations were depressed by repeated HGF gene transfer. It is known that the P2X3 receptor is expressed by unmyelinated sensory nerve fibers and mediates thermal hyperalgesia, whereas P2Y1 receptors are expressed by myelinated sensory nerve fibers and mediate mechanical allodynia [58–60]. TRPV1 is essential for tissue injury-induced thermal hyperalgesia [61]. Our results suggest when HGF protein is induced via nonviral retrograde gene transfer, it may act directly on the nervous system to exert effects leading to improvements in mechanical and thermal thresholds [16,22–25,42]. Intriguingly, we failed to find increases in P2X2 or P2X4 mRNA levels after IRI. P2X2/3 heteromerics show a relation to mechanical allodynia induction [62]. Tsuda et al. reported that mechanical allodynia requires activation of the P2X4 receptor, and nerve injury-induced pain depends upon ongoing signaling via the P2X4 receptor [59]. Indeed, in our previous study using a rat model of CCI, we demonstrated preventive effect of repeated HGF gene transfer on an induction of mechanical allodynia and elevations in the levels of P2X2, P2X4, and P2Y1 receptor mRNAs [29].

## Conclusion

We have demonstrated that transfer of the human HGF gene into the rat nervous system by repeated intramuscular injection of HVJ envelope resulted in potent angiogenesis and subsequently enhanced nerve regeneration in ischemic nerves. These results support HGF transfer as a promising candidate for treatment of acute ischemic nerve injury caused by reperfusion injury.

## Supporting information

**S1 Table. Primer sequences, TaqMan probe sequences, and thermocycle conditions used for RT-PCR.**
(DOCX)

**S2 Table. Raw data of withdrawal threshold (g) using Von Frey hairs of bilateral hind paws in IRI+HGF, IRI, control and sham groups.**
(DOCX)

**S3 Table. Morphometric data at the lower-thigh level of bilateral sciatic nerves at 3 weeks post-surgery: Fascicular area, and the density and total number of endoneurial microvessels in IRI+HGF, IRI, control and sham groups.**
(DOCX)

**S4 Table. Morphometric data at the lower-calf level of bilateral tibial nerves at 3 weeks post-surgery: Fascicular area, and the density of endoneurial microvessels in IRI+HGF, IRI, and control groups.**
(DOCX)

**S1 Fig.** Neurometer measurements of stimulus threshold at 5 **(A)**, 250 **(B)** and 2,000 **(C)** Hz to the plantar surface of the left hind paw in IRI (open bars) and IRI+HGF (solid bars) groups, and pre-surgical levels (Pre OP, checkered bars). Data are expressed as mean ± SD (n = 6 for each group at each time point). *$p < 0.01$: IRI *vs*. IRI+HGF.
(TIF)

**S2 Fig. Electron micrograph at the lower-thigh level of right sciatic nerve at 3 weeks after surgery showing normal unmyelinated nerve fibers.** Note myelinated nerve fibers with disproportionally thin myelin relative to axon. area Bar = 2μm.
(TIF)

**S3 Fig. Histograms of axon diameter against axon numbers of myelinated nerve fibers at lower-thigh level of right sciatic nerves at 3 weeks post-surgery in IRI (dotted line), IRI +HGF (solid line), and control (broken line) groups.**
(TIF)

**S4 Fig. The total number of endoneurial microvessels at the lower-thigh level of sciatic nerves 3 weeks after surgery in IRI+HGF (closed bars), IRI (open bars), control (checkered bar) and sham (dotted bar) groups.** *$p < 0.01$: IRI+HGF *vs*. IRI.
(TIF)

**S5 Fig.** Semi-thin transverse sections at the upper thigh level of the contralateral (left) sciatic nerves at 3 weeks post-surgery showing endoneurial microvessels (arrows) in IRI+HGF (A) and IRI (B) groups. The number of endoneurial microvessels in HGF-treated nerve (A) was apparently greater than in IRI nerve (B). Bars = 25μm. *perineurium.
(TIF)

**S6 Fig. Fascicular area and the density of endineurial microvessels at the lower-calf level of bilateral tibial nerves 4 weeks after surgery in IRI+HGF (closed bars), IRI (open bars) and control (checkered bar) groups.**
(TIF)

**S7 Fig. RT-PCR data of DGR: mRNA expressions of P2X2, P2X3, P2X4, and P2Y1 receptors, and TRPV1 at weeks 1 and 3 post-surgery in IRI (open bars), IRI+HGF (solid bars), and control (checkered bars) groups.** Data are expressed as mean ± SD (n = 6 for each group at each time point).
(TIF)

## Author Contributions

**Conceptualization:** Toyokazu Tsuchihara, Kuniaki Nakanishi, Masatoshi Amako, Hiroshi Arino, Koichi Nemoto.

**Data curation:** Toyokazu Tsuchihara, Hitoshi Nukada, Kuniaki Nakanishi.

**Formal analysis:** Toyokazu Tsuchihara, Hitoshi Nukada, Kuniaki Nakanishi, Masatoshi Amako, Hiroshi Arino, Koichi Nemoto.

**Funding acquisition:** Toyokazu Tsuchihara, Koichi Nemoto.

**Investigation:** Toyokazu Tsuchihara, Hitoshi Nukada, Kuniaki Nakanishi.

**Methodology:** Toyokazu Tsuchihara, Hitoshi Nukada, Kuniaki Nakanishi, Ryuichi Morishita, Masatoshi Amako.

**Project administration:** Toyokazu Tsuchihara, Hitoshi Nukada, Kuniaki Nakanishi, Masatoshi Amako, Hiroshi Arino, Koichi Nemoto.

**Resources:** Toyokazu Tsuchihara, Hitoshi Nukada, Kuniaki Nakanishi.

**Supervision:** Kuniaki Nakanishi, Ryuichi Morishita, Masatoshi Amako, Hiroshi Arino, Koichi Nemoto, Kazuhiro Chiba.

**Validation:** Toyokazu Tsuchihara, Hitoshi Nukada, Kuniaki Nakanishi, Hiroshi Arino, Koichi Nemoto.

**Visualization:** Toyokazu Tsuchihara, Hitoshi Nukada, Kuniaki Nakanishi, Masatoshi Amako.

**Writing – original draft:** Toyokazu Tsuchihara, Hitoshi Nukada.

**Writing – review & editing:** Toyokazu Tsuchihara, Hitoshi Nukada, Kuniaki Nakanishi, Ryuichi Morishita, Masatoshi Amako, Hiroshi Arino, Koichi Nemoto, Kazuhiro Chiba.

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
