## [Decision Letter · Decision Letter 0]

24 Apr 2020

PONE-D-20-09804

Efficacy of nonviral gene transfer of human hepatocyte growth factor (HGF) against ischemic-reperfusion nerve injury in rats

PLOS ONE

Dear Dr. Nukada,

Thank you for submitting your manuscript to PLOS ONE. After careful consideration, we feel that it has merit but does not fully meet PLOS ONE’s publication criteria as it currently stands. Therefore, we invite you to submit a revised version of the manuscript that addresses the points raised during the review process.

We would appreciate receiving your revised manuscript by Jun 08 2020 11:59PM. To enhance the reproducibility of your results, we recommend that if applicable you deposit your laboratory protocols in protocols.io, where a protocol can be assigned its own identifier (DOI) such that it can be cited independently in the future. For instructions see: http://journals.plos.org/plosone/s/submission-guidelines#loc-laboratory-protocols

We look forward to receiving your revised manuscript.

Kind regards,

Soroku Yagihashi, MD, PhD

Academic Editor

PLOS ONE

Additional Editor Comments (if provided):

The submitted paper was sent to the external reviewers specialized in this field. Both reviewers admitted the value of this study worthy publication. However, they also pointed out several issues that should be carefully attended to before the acceptance for publication.

From the handling editor, some additional issues are suggested for the improvement of the content.

1. The mechanism of how HGF worked for the correction of ischemia/reperfusion injury is not still clear. Confirmation of HGF transfer to peripheral nervous system was not well evaluated. Thus, the statement in the text should be toned down.

2. Regarding the above issue, the authors evaluated only ipsilateral side of surgical procedure. There is no statement on the changes of contralateral side, nor any statement on the effects of HGF on this side. If there occurs some improvement in rats with HGF treatment in opposite side, correction of vascular impairement may be implicated for the improvement.

3. Morphometry of endoneurial as well as epineurial vasculature may be worth for the improvement of content referring implication of endoneurial edema and its improvement by HGF.

4. Evidence of promoted nerve fiber regeneration in HGF-treatment group is obscure. Pictures provided are not sufficient for recognition of degenerated or regenerated fibers. There appears to be endoneurial edema which should be evaluated by morphometry. Was it possible to consider that the data of fiber (axonal) atrophy in ischemic neuropathy were rescued by HGF?

5. Figure 1; IRI+HGF should be open square, not closed

6. Angiogenetic should be angiogenic.

2. To comply with PLOS ONE submissions requirements, in your Methods section, please provide additional information on the animal research and ensure you have included details on (1) methods of sacrifice, (2) methods of analgesia, and (3) efforts to alleviate suffering.

Reviewers' comments:

Reviewer's Responses to Questions

**Comments to the Author**

1. Is the manuscript technically sound, and do the data support the conclusions?

Reviewer #1: Yes

Reviewer #2: Yes

2. Has the statistical analysis been performed appropriately and rigorously? 

Reviewer #1: Yes

Reviewer #2: No

3. Have the authors made all data underlying the findings in their manuscript fully available?

Reviewer #1: No

Reviewer #2: Yes

4. Is the manuscript presented in an intelligible fashion and written in standard English?

Reviewer #1: Yes

Reviewer #2: Yes

5. Review Comments to the Author

Reviewer #1: This is a detailed series of experiments employing behavioural, functional and morphometric and molecular techniques to evaluate the benefit of HGF gene transfer after ischemia reperfusion injury.

The experimental protocols and methods are appropriate.

The results are of interest and build on previously published data showing HGF efficacy in STZ diabetic neuropathy and in a small clinical trial of patients with painful neuropathy (Kessler et al Ann Clin Transl Neurol 2015; 2: 465-478).

However, I believe the translational value of this approach in ischemic neuropathy is potentially limited as patients often have long periods of chronic ischemia before an acute occlusion may develop.

The conclusion is focused on the beneficial effect of HGF on ‘symptoms’

‘resulted in a reducing effect on symptoms of ischemic neuropathy caused by IRI’.

However, apart from mechanical allodynia which is an indirect assessment of symptoms, no other symptoms have been evaluated in this model?

The pain is principally mediated by small fibres, yet the morphological examination has been undertaken in large myelinated fibres?

The effect on the vasculature which is the mechanism by which HGF is assumed to work has either not been quantified or is very qualitatively presented. Was there any change in endoneurial or epineurial vascular density or morphology (luminal size, endothelial cell number/area)?

Is motor function altered in these animals after IRI?

Why is MNCV slower in the IRI+HGF group?

Should myelinated nerve fibre regeneration not be associated with a decrease in axon and fibre areas due to the presence of regenerating fibres?

There is no morphological examination of the C-fibres, either with EM in the sciatic nerve or IENFD?

What was the effect on fascicular area as a measure of endoneurial oedema?

Is HGF angiogenetic or angiogenic?

Line 297 area not are

Reviewer #2: Tsuchihara et al. reported the potential efficacy of non-viral human HGF gene transfer toward ischemic neuropathy in rats. In this study, HGF gene transfer alleviated mechanical allodynia and thermal hyperalgesia, and restored skin blood flow and temperature, neurometer current perception threshold, and motor nerve conduction velocity and compound action potential amplitude in rats with ischemic-reperfusion nerve injury (IRI). In addition, HGF gene transfer ameliorated the reduced axon diameter and area of myelinated nerve fibers and the up-regulation of P2X3 and TRPV1 receptor mRNA expression in sciatic nerves in IRI rats.

The authors employed various kinds of techniques to evaluate large and small nerve fiber function and morphometry in IRI rats. The manuscript is basically comprehensible, the purpose of the study is clear, and the findings are straightforward. Using the similar methodology, the authors demonstrated the efficacy of transient expression of HGF toward CCI and diabetic neuropathy in rats. In addition to those studies, the present study indicates usefulness of HGF gene transfer toward ischemic neuropathy. However, several drawbacks of this manuscript need to be pointed out.

1. The authors failed to provide evidence of successful HGF gene transfer into IRI rats.

2. The authors stated that there were no significant changes between the [control] and [sham] groups (Lines 196-197, 204-205, 210-211), but failed to provide data. They need to show the data as supplementary information.

3. In Figs.1-4, no statistical analyses were performed between [control] and [IRI] or [IRI+HGF]. It is unclear why the authors did not use [sham] for comparison with [IRI] or [IRI+HGF].

4. It is unclear which scale indicates 1-8 weeks in the horizontal axis of Fig.1A and B.

5. A representative photomicrograph for [control] or [sham] should be added to Fig.5A. It is also important to indicate the effects of HGF gene transfer on the morphology of blood vessels and/or endothelial cells in IRI rats.

6. In Fig.6, the authors conducted classical semiquantitative RT-PCR analysis; currently, real-time RT-PCR is a standard methodology. If the authors want to use these data, it would be recommended to show them as supplementary information and tone down the statement based on these findings.

6. PLOS authors have the option to publish the peer review history of their article (what does this mean?). If published, this will include your full peer review and any attached files.

Reviewer #1: Yes: Rayaz A Malik

Reviewer #2: No

---

## [Author Response · Author response to Decision Letter 0]

15 Jul 2020

Additional Editor Comments (if provided):

1. The mechanism of how HGF worked for the correction of ischemia/reperfusion injury is not still clear. Confirmation of HGF transfer to peripheral nervous system was not well evaluated. Thus, the statement in the text should be toned down.

The reviewer is correct. In the current study, we have not demonstrated the exact molecular mechanism of HGF on ischemic /reperfusion injury. 

However, we have confirmed the HGF transfer into the PNS in our previous studies using exactly the same technique. New sentences have been added: Line 69-70, Line 293-307. 

2. Regarding the above issue, the authors evaluated only ipsilateral side of surgical procedure. There is no statement on the changes of contralateral side, nor any statement on the effects of HGF on this side. If there occurs some improvement in rats with HGF treatment in opposite side, correction of vascular impairment may be implicated for the improvement.

Data from the contralateral side was included. 

Following sentences, figures and tables have been replaced or added: 

Line 196-9, 210-1, 214-6, 222-5, 327-35.

Figs: 2, 3 & 7, S Figs: 1, 4, 5 & 6.

S Tables: 2, 3 & 4.

3. Morphometry of endoneurial as well as epineurial vasculature may be worth for the improvement of content referring implication of endoneurial edema and its improvement by HGF.

Yes, we have added the morphometric analysis of endoneurial microvessels. In this study, all endoneurial vessels which diameter less than 30 μm were counted as microvessels. 

Following sentences, figs and tables have been added:

Line 48, 50, 172, 253-60, 288-9, 319-21, Fig 7, S Figs 4, 5 & 6, and S Tables 3 & 4. 

4. Evidence of promoted nerve fiber regeneration in HGF-treatment group is obscure. Pictures provided are not sufficient for recognition of degenerated or regenerated fibers. There appears to be endoneurial edema which should be evaluated by morphometry. Was it possible to consider that the data of fiber (axonal) atrophy in ischemic neuropathy were rescued by HGF?

Fig 6 was revised. 

An EM photo was added (S2 Fig). 

Fascicular area was measured (Fig 7, S6 Fig, S Tables 3 & 4)).

5. Figure 1; IRI+HGF should be open square, not closed

Fixed.

6. Angiogenetic should be angiogenic.

Corrected..

Reviewer #1: This is a detailed series of experiments employing behavioural, functional and morphometric and molecular techniques to evaluate the benefit of HGF gene transfer after ischemia reperfusion injury.

The experimental protocols and methods are appropriate.

The results are of interest and build on previously published data showing HGF efficacy in STZ diabetic neuropathy and in a small clinical trial of patients with painful neuropathy (Kessler et al Ann Clin Transl Neurol 2015; 2: 465-478).

Thanks so much. We have quoted the above paper by Kessler et al. 2015 (Line 305-7).

However, I believe the translational value of this approach in ischemic neuropathy is potentially limited as patients often have long periods of chronic ischemia before an acute occlusion may develop.

The conclusion is focused on the beneficial effect of HGF on ‘symptoms’

‘resulted in a reducing effect on symptoms of ischemic neuropathy caused by IRI’.

However, apart from mechanical allodynia which is an indirect assessment of symptoms, no other symptoms have been evaluated in this model?

The reviewer is correct. We have used the model of acute ischemia, and chronic ischemia is another story. The word ‘acute’ has been added (Line54 & 356). 

We have also evaluated thermal and Neurometer thresholds, although not sensory symptoms. However, the word ‘symptoms’ has been deleted in the conclusion.

The pain is principally mediated by small fibres, yet the morphological examination has been undertaken in large myelinated fibres?

We have added a picture of unmyelinated nerve fibers (S2 Fig), and a new sentence (Line 242-3). 

The effect on the vasculature which is the mechanism by which HGF is assumed to work has either not been quantified or is very qualitatively presented. Was there any change in endoneurial or epineurial vascular density or morphology (luminal size, endothelial cell number/area)?

Vascular morphometry of endoneurials has been added as mentioned above. There are no apparent abnormalities of vascular morphology (Line 243-4).

Is motor function altered in these animals after IRI?

Why is MNCV slower in the IRI+HGF group?

Yes, foot/toe drop has developed in ischemic limbs, and we have added a sentence (Line 233).

Even rats in HGF group showed nerve degeneration including large myelinated nerve fibers, but the recovery is significantly faster than non HGF-treated nerves. .

Should myelinated nerve fibre regeneration not be associated with a decrease in axon and fibre areas due to the presence of regenerating fibres?

Yes, we entirely agree with the reviewer. Morphometry was done after 3 weeks of surgery only in the current study. As the reviewer suggested, we added new sentences (Line 323-6). 

There is no morphological examination of the C-fibres, either with EM in the sciatic nerve or IENFD?

We have added an EM photo of unmyelinated nerve fibers (S2 Fig).

What was the effect on fascicular area as a measure of endoneurial oedema?

We have added these data (Fig 7, S6 Fig, S Table 3 & 4))

Is HGF angiogenetic or angiogenic?

Angiogenic. Fixed.

Line 297 area not are

Corrected (now Line 321). 

Reviewer #2: Tsuchihara et al. reported the potential efficacy of non-viral human HGF gene transfer toward ischemic neuropathy in rats. In this study, HGF gene transfer alleviated mechanical allodynia and thermal hyperalgesia, and restored skin blood flow and temperature, neurometer current perception threshold, and motor nerve conduction velocity and compound action potential amplitude in rats with ischemic-reperfusion nerve injury (IRI). In addition, HGF gene transfer ameliorated the reduced axon diameter and area of myelinated nerve fibers and the up-regulation of P2X3 and TRPV1 receptor mRNA expression in sciatic nerves in IRI rats.

The authors employed various kinds of techniques to evaluate large and small nerve fiber function and morphometry in IRI rats. The manuscript is basically comprehensible, the purpose of the study is clear, and the findings are straightforward. Using the similar methodology, the authors demonstrated the efficacy of transient expression of HGF toward CCI and diabetic neuropathy in rats. In addition to those studies, the present study indicates usefulness of HGF gene transfer toward ischemic neuropathy. However, several drawbacks of this manuscript need to be pointed out.

1. The authors failed to provide evidence of successful HGF gene transfer into IRI rats.

Please refer the comment 1 to the editor.

2. The authors stated that there were no significant changes between the [control] and [sham] groups (Lines 196-197, 204-205, 210-211), but failed to provide data. They need to show the data as supplementary information.

Yes, we have added the data of sham group (Fig 7, S4 & 6 Figs , and S2, 3 & 4 Tables). 

3. In Figs.1-4, no statistical analyses were performed between [control] and [IRI] or [IRI+HGF]. It is unclear why the authors did not use [sham] for comparison with [IRI] or [IRI+HGF].

As mentioned in the text, there is no significant difference between control and sham groups. Because of the simplicity of figures, we have shown only three limes in Figs 1A & 1B. 

Because focusing on the effect of HGF, we emphasized data of IRI+HGF and IRI groups. 

4. It is unclear which scale indicates 1-8 weeks in the horizontal axis of Fig.1A and B.

Figs 1A and 1B were revised. 

5. A representative photomicrograph for [control] or [sham] should be added to Fig.5A. It is also important to indicate the effects of HGF gene transfer on the morphology of blood vessels and/or endothelial cells in IRI rats.

A photo of sham group was added in Fig 6.

Morphometry of endoneurial microvessels was added as mentioned above.

6. In Fig.6, the authors conducted classical semiquantitative RT-PCR analysis; currently, real-time RT-PCR is a standard methodology. If the authors want to use these data, it would be recommended to show them as supplementary information and tone down the statement based on these findings.

A part of Fig 8 (originally Fig 6) was moved to S7 Fig.

---

## [Decision Letter · Decision Letter 1]

22 Jul 2020

Efficacy of nonviral gene transfer of human hepatocyte growth factor (HGF) against ischemic-reperfusion nerve injury in rats

PONE-D-20-09804R1

Dear Dr. Nukada,

We’re pleased to inform you that your manuscript has been judged scientifically suitable for publication and will be formally accepted for publication once it meets all outstanding technical requirements.

Kind regards,

Soroku Yagihashi, MD, PhD

Academic Editor

PLOS ONE

Additional Editor Comments (optional):

The authors well responded to the reviewers' and my comments. The revised version has much been improved and well readable.

Reviewers' comments:

Reviewer's Responses to Questions

**Comments to the Author**

1. If the authors have adequately addressed your comments raised in a previous round of review and you feel that this manuscript is now acceptable for publication, you may indicate that here to bypass the “Comments to the Author” section, enter your conflict of interest statement in the “Confidential to Editor” section, and submit your "Accept" recommendation.

Reviewer #1: All comments have been addressed

Reviewer #2: All comments have been addressed

2. Is the manuscript technically sound, and do the data support the conclusions?

Reviewer #1: Yes

Reviewer #2: Yes

3. Has the statistical analysis been performed appropriately and rigorously? 

Reviewer #1: Yes

Reviewer #2: Yes

4. Have the authors made all data underlying the findings in their manuscript fully available?

Reviewer #1: No

Reviewer #2: Yes

5. Is the manuscript presented in an intelligible fashion and written in standard English?

Reviewer #1: Yes

Reviewer #2: Yes

6. Review Comments to the Author

Reviewer #1: All my comments have been addressed and the authors have made changes to address my concerns.

Although the fact that unmyelinated nerve fibres and vessels are not abnormal does beg the question as to the underlying pathophysiology of this model.

Reviewer #2: The manuscript has been revised in an appropriate way, and the authors provided additional data and figures. I have no further comments on it.

7. PLOS authors have the option to publish the peer review history of their article (what does this mean?). If published, this will include your full peer review and any attached files.

Reviewer #1: **Yes: **Rayaz Malik

Reviewer #2: No

---

## [Editor Report · Acceptance letter]

24 Jul 2020

PONE-D-20-09804R1 

Efficacy of nonviral gene transfer of human hepatocyte growth factor (HGF) against ischemic-reperfusion nerve injury in rats 

Dear Dr. Nukada:

I'm pleased to inform you that your manuscript has been deemed suitable for publication in PLOS ONE. Congratulations! Your manuscript is now with our production department. 

Kind regards, 

on behalf of

Professor Soroku Yagihashi 

Academic Editor

PLOS ONE